# Estimating genetic variance contributed by a quantitative trait locus: A random model approach

**Shibo Wang**◉[º], **Fangjie Xie**[º], **Shizhong Xu**◉*

Department of Botany and Plant Sciences, University of California, Riverside, California, United States of America

◉ These authors contributed equally to this work.
* shizhong.xu@ucr.edu

## Abstract

Detecting quantitative trait loci (QTL) and estimating QTL variances (represented by the squared QTL effects) are two main goals of QTL mapping and genome-wide association studies (GWAS). However, there are issues associated with estimated QTL variances and such issues have not attracted much attention from the QTL mapping community. Estimated QTL variances are usually biased upwards due to estimation being associated with significance tests. The phenomenon is called the Beavis effect. However, estimated variances of QTL without significance tests can also be biased upwards, which cannot be explained by the Beavis effect; rather, this bias is due to the fact that QTL variances are often estimated as the squares of the estimated QTL effects. The parameters are the QTL effects and the estimated QTL variances are obtained by squaring the estimated QTL effects. This square transformation failed to incorporate the errors of estimated QTL effects into the transformation. The consequence is biases in estimated QTL variances. To correct the biases, we can either reformulate the QTL model by treating the QTL effect as random and directly estimate the QTL variance (as a variance component) or adjust the bias by taking into account the error of the estimated QTL effect. A moment method of estimation has been proposed to correct the bias. The method has been validated via Monte Carlo simulation studies. The method has been applied to QTL mapping for the 10-week-body-weight trait from an $F_2$ mouse population.

## Author summary

One of the goals of QTL mapping and GWAS is to quantify the size of a QTL, which is measured by the QTL variance or the proportion of trait variance explained by the QTL. The effect of a QTL appears in a linear or linear mixed model as a regression coefficient and defined as a fixed effect. The estimated QTL variance in conventional QTL mapping studies takes the square of the estimated QTL effect. This is a biased estimate of QTL variance. An unbiased estimate of the QTL variance should be obtained by (1) treating the QTL effect as random and estimating the variance of the random effect or (2) adjusting

**Data Availability Statement:** All relevant data are within the manuscript and its Supporting Information files.

**Funding:** The project was partially supported by the Hatch Project CA-R-BPS-5028-H to SX. The

funders had no role in study design, data collection and analysis, decision to publish, or preparation of the manuscript.

**Competing interests:** The authors have declared that no competing interests exist.

the squared estimated QTL effect by the squared estimation error. We proved that the two methods are identical. We further proved that the usual $R^2$ (goodness of fit) in regression analysis is equivalent to the biased QTL heritability while the adjusted $R^2$ is equivalent to the bias corrected QTL heritability.

This is a *PLOS Computational Biology* Methods paper.

# 1 Introduction

Quantitative trait locus (QTL) mapping [1] and genome-wide association studies (GWAS) [2] are the main tools for identifying genomic regions harboring quantitative trait loci. These QTL regions are the targets for molecular geneticists to further expand the experiments, to clone the actual genes for agronomic traits and to help breeders develop optimal marker assisted selection (MAS) programs [3]. Goring et al. [4] stated that the primary goal of QTL mapping and GWAS is to locate QTL and the secondary goal is to quantify the sizes of QTL. The size of a QTL is represented by the squared QTL effect or the QTL variance. We believe that estimating the variances of QTL is equally important as locating the QTL because only QTL with large effects are useful for application while small effect but statistically significant QTL are not economically meaningful. Statistical significance is primarily determined by the sample size. A small-effect QTL can be detected in a very large sample, but such a small-effect QTL is useless in any breeding programs. The final reported variance of a detected QTL is often converted into the proportion of phenotypic variance contributed by the QTL, called the QTL heritability [5,6]. In addition, whether a QTL is large or small is determined relative to the residual or phenotypic variance.

In interval mapping [1], composite interval mapping [7,8] and genome-wide association studies [2], the effect of a QTL appears as a regression coefficient in a linear model or a linear mixed model. The regression coefficient is a parameter in the model. The least squares or maximum likelihood estimate of a QTL effect is often unbiased [1,9]. However, when the unbiased estimate of the QTL effect is converted into a squared QTL effect, i.e., QTL variance, the estimated variance is no longer unbiased. The bias can be substantially high for small-effect QTL detected from small samples [6]. This bias is not related to the Beavis effect, which is primarily caused by significance tests [10–12]. Biased estimates of QTL variances discussed in previous literature is almost all related to significance tests, i.e., the Beavis effect. However, the bias can occur even if there is no significance test associated with the estimation and this bias has been virtually ignored in the QTL mapping community [6]. There was little theoretical explanation for the bias. Broman [13] and Allison et al. [14] also noted that estimates of non-significant QTL effects may also be biased, primarily due to the constraints of QTL parameters. For example, if QTL heritability is the parameter, its solution space must be constrained between 0 and 1. If the true QTL heritability is close to 0 or close to 1, then the estimated QTL heritability will be biased towards the middle of the constrained interval.

Beavis effect is a phenomenon that reported QTL from relatively small samples are often larger than they actually are [4,10–12]. The current study is focused on bias in estimated QTL variances not due to the Beavis effect but due to a wrong statistical model being used.

The current models for QTL mapping and genome-wide association studies are linear models and linear mixed models [2,8,15,16]. The effect of a QTL appears in these models as a

parameter that is subject to estimation. The QTL variance is often defined as the squared QTL effect and the estimated QTL variance is simply obtained by squaring the estimated QTL effect [5,6]. In general, the QTL variance is determined by the QTL effect and the frequency of the QTL alleles in the target population [17]. For example, in an $F_2$ design of QTL mapping, suppose that we code the genotype indicator variable as $Z = \sqrt{2}$ for $A_1A_1$, $Z = 0$ for $A_1A_2$ and $Z = -\sqrt{2}$ for $A_2A_2$ [18]. Let $\alpha$ be the average effect of gene substitution, i.e., the QTL effect [19]. In the absence of dominance and no segregation distortion, the QTL variance is defined as $\sigma^2_{QTL} = \mathrm{var}(Z)\alpha^2 = \alpha^2$, where $\mathrm{var}(Z) = \sigma^2_Z = \mathrm{E}(Z^2) - \mathrm{E}^2(Z) = 1 - 0 = 1$.

In classical quantitative genetics [19,20], the effect of a quantitative trait locus is treated as a fixed effect but the genotype indicator variable is treated as random. The QTL variance is defined as $\sigma^2_{QTL} = \sigma^2_Z\alpha^2 = 2pq\alpha^2$, where $p$ is the frequency of the "high" allele and $q = 1–p$ is the frequency of the "low" allele and $\sigma^2_Z = 2pq$ is the variance of the genotype indicator variable under the assumption of Hardy-Weinberg equilibrium. The genotype indicator variable $Z$ here is coded as the number of "high" alleles in one of the three genotypes, i.e., $Z_j = 2$ for genotype $A_1A_1$, $Z_j = 1$ for genotype $A_1A_2$ and $Z_j = 0$ for genotype $A_2A_2$. The textbook [19] provides a model for the variance and no estimation of the variance is presented. When the dominance effect is absent ($d = 0$) or the two alleles have an equal frequency, the average effect of gene substitution is defined as $\alpha = a + d(q–p) = a$, where $a = G_{11} - \frac{1}{2}(G_{11} + G_{22})$ is called the "additive effect." The genotypic value ($G_{11}$) is interpreted as the average trait value from all individuals with genotype $A_1A_1$ and $G_{22}$ is the average trait value for all individuals with genotype $A_2A_2$. The genotypic values ($G_{11}$ and $G_{22}$) are not estimated parameters from a finite sample but the true genotypic values under the assumption of being obtained from an infinitely large sample. In classroom teaching, an instructor may use a finite sample to demonstrate how $G_{ij}$ is obtained, but the genotypic values are defined as the true values. A naïve estimate of the additive variance is $\hat{\sigma}^2_{QTL} = 2pq\hat{\alpha}^2$, which is an over estimate of the additive variance.

The statistical models of QTL mapping in a designed experiment are fixed models because the QTL effect ($\alpha$) is the parameter subject to estimation and no distribution is assigned to this fixed effect. We are not criticizing the fixed effect models in classical quantitative genetics; rather, we point out that the naïve estimate of the QTL variance (estimated effect squared) is biased. Gianola et al [21] first systematically investigated the properties of this QTL variance. They assigned a normal distribution to $\alpha$, where the variance of that distribution is interpreted as a prior uncertainty in the Bayesian framework. Chen et al [22] also proposed to treat the QTL effect as random and the QTL variance as the parameter. In fact, Chen et al [22] investigated the problem from an empirical Bayes point of view so that the QTL variance is the parameter of the prior distribution of the QTL effect. More detailed analysis of QTL variance can be found in Gianola et al. [23]. If we treat the QTL variance as the parameter of interest and directly estimate the QTL variance, the bias will disappear. The first random model approach to QTL mapping was proposed by Fernando and Grossman [24] for pedigree data analysis followed by the random model interval mapping developed by Xu and Atchley [25] for sib family data analysis. The repeated $F_2$ design of experiment [17,26] is an extension of the simple $F_2$ design of experiment initiated by crossing a common parent with multiple independent inbred lines. Since multiple parents are involved, the effects of parental alleles are treated as random effects with mean zero and an unknown variance. This variance is the QTL variance, which can be estimated via the maximum likelihood method. The QTL variance is tested with a likelihood ratio test [17,26]. QTL variance estimated this way is asymptotically unbiased or with little bias in finite samples. More random model QTL mapping procedures were developed in a short span of a half dozen years towards the end of the 20th century [27,28]. When the QTL effect is treated as a random effect, the parameter is the QTL variance and thus no

bias or little bias is expected for the estimated QTL variance. Therefore, treating the QTL effect as a random effect and estimating the variance of the random effect is an alternative way of estimating QTL variance. We call the models with QTL effects treated as random effects random models, although they can be mixed models, technically, because a random polygenic effect may be included in the models. Note that we are talking about the bias in an estimated QTL variance, regardless whether the QTL is statistically significant or not. The sib-pair regression analysis of QTL mapping [29–32] is a fixed model (not a random model), but the parameter itself (regression coefficient) is already the QTL variance and thus there is no bias associated with the QTL variance in sib-pair regression analysis. The bottom line is that if the parameter subject to estimation is already the QTL variance, no bias or little bias is expected other than the bias caused by the Beavis effect.

Some genomic selection models have been adopted for multiple locus GWAS, e.g., models of the Bayesian alphabet for genomic selection [21,33–36]. In Bayes A, markers of the entire genome are included in a single model. Because the number of markers can be substantially larger than the sample size, each marker effect is assigned a normal prior with mean zero and an unknown variance. The prior variance of each marker is further described by a hyper prior distribution so that the marker variance can be obtained via the posterior mean or posterior mode estimation [34,37–40]. Marker variances obtained this way are not biased because they are directly estimated from the data, not converted from the squared marker effects. In Bayes B, each marker effect is assigned a mixture of two distributions, one is a normal distribution and the other is just a zero with some non-zero probability mass [34,41]. The variance of the normal distribution in the mixture is the marker variance. This variance is also unbiased or with very little bias because the variance is not converted from the squared marker effect.

In contrast to the Bayesian alphabetic series of genomic selection models, the genomic best linear unbiased prediction (GBLUP) [42], which is the same as the ridge regression [43,44], cannot be used for GWAS in its original form because all markers are assigned to the same normal distribution. The single variance is shared by all markers and is severely shrunk towards zero. However, the test statistic of each marker from the ridge regression can be de-shrunk to a comparable level as the typical mixed model GWAS [45–47]. Duarte et al. [45] de-shrank the test for each marker so that the Wald test statistic was brought back to a level similar to the test of EMMA [15,48]. However, Duarte et al. [45] only de-shrank the test and the estimated effect for each marker remains the same as the ridge regression. The two-step ridge regression approach to GWAS developed by Shen et al [46] de-shrank both the effects and the tests. The de-shrunk marker variances may be used to calculate the QTL heritability. Wang and Xu (2020) recently developed another de-shrinking method that can de-shrink both the test, the estimated marker effect, and the estimated marker variance. This variance is unbiased and can be directly used to calculate the QTL heritability. The methods summarized here are various extensions from the genomic selection models. They are not the typical methods of GWAS. The typical methods are represented by EMMA and GEMMA [15,49].

An unbiased estimate of QTL variance will lead to a less biased estimate of QTL heritability, which is expressed by $\hat{h}_{QTL}^2 = \hat{\sigma}_{QTL}^2 / (\hat{\sigma}_{QTL}^2 + \hat{\sigma}^2)$, where $\hat{\sigma}^2$ is the estimated residual variance. The QTL heritability has many different definitions, (1) proportion of the phenotypic variance contributed by the QTL variance, (2) R squared, which is defined as the ratio of the regression sum of squares to the total sum of squares, (3) Adjusted $R^2$, which is a modified $R^2$ by accounting for the number of independent variables, (4) pseudo $R^2$ [50,51], which is designed for logistic regression analysis for binary traits. All the $R^2$ related measurements, except the adjusted $R^2$, may be called the model goodness of fit. We will show in the discussion that the model goodness of fit is a biased estimate of the QTL heritability.

The purpose of this study is to investigate the bias in the estimated QTL variance when the QTL effect is treated as a fixed effect (in plants). We show that the bias disappears when the QTL effect is treated as a random effect. We also propose a moment method to correct the bias. The bias due to significance test (the Beavis effect) has been investigated by our laboratory in a recent study where a truncated non-central Chi-square distribution has been used to derive and correct the bias [52]. This study only focuses the bias due to the use of an incorrect statistical model. We emphasize more on the conceptual issue than the practical application issue.

## 2 Method

### 2.1 Model of a quantitative trait locus

Let $y$ be a vector of phenotypic values for a quantitative trait collected from a mapping population. The trait value can be described by the following linear mixed model,

$$y = X\beta + Z\alpha + \xi + \varepsilon \tag{1}$$

where $X\beta$ represents fixed effects not associated with genes. If there are no fixed effects other than the population mean, $X\beta = 1\mu$, where $X = 1$ is a column vector of unity and $\beta = \mu$ is the population mean (or intercept). Let $g = Z\alpha$ be an $n \times 1$ vector of genotypic values for all individuals. The model is rewritten as

$$y = 1\mu + g + \xi + \varepsilon \tag{2}$$

where

$\mu$ is the population mean,

$g$ is a vector of genotypic values for all individuals,

$\xi$ is a vector of polygenic effects with an assumed $N\left(0, A\sigma_\xi^2\right)$ distribution,

$A$ is an additive relationship matrix, also called numerator matrix,

$\sigma_\xi^2$ is a polygenic variance,

$\varepsilon$ is a vector of residual errors with an assumed $N(0, R\sigma^2)$ distribution,

$R$ is a residual covariance structure (often assumed to be $R = I$),

$\sigma^2$ is the residual variance.

Let $Z_j$ be the genotype indicator variable of individual $j$ for the locus of interest, which is defined as

$$Z_j = \begin{cases} +1 & \text{for } A_1A_1 \text{ with } P = p^2 \\ 0 & \text{for } A_1A_2 \text{ with } H = 2pq \\ -1 & \text{for } A_2A_2 \text{ with } Q = q^2 \end{cases} \tag{3}$$

where $p = \Pr(A_1)$ is the frequency of allele $A_1$ and $q = \Pr(A_2)$ is the frequency of allele $A_2$, where $p + q = 1$. The three capital letters, $P$, $H$ and $Q$ are the frequencies of the three genotypes and the population is assumed to be in Hardy-Weinberg equilibrium. Let $\alpha$ be the genetic effect of the locus, which is often called the average effect of gene substitution in classical quantitative genetics textbooks [19,20]. Since there is no distribution assigned to the QTL effect $\alpha$, it is a fixed effect. The genetic variance contributed by the locus under the

fixed model is defined as

$$\sigma_{QTL}^2 = \sigma_Z^2 \alpha^2 = 2pq\alpha^2 \tag{4}$$

Here, the genetic effect $\alpha$ is a fixed effect (constant) and $Z$ is a random variable with mean $\mu_Z = p-q$ and variance $\sigma_Z^2 = 2pq$. Although $\alpha$ is fixed, $g = Z\alpha$ is random because $Z$ is random. The Hardy-Weinberg equilibrium assumption is not required and we made that assumption here is to be consistent with the classical definition of genetic variance defined in quantitative genetics textbooks [19,20].

When $Z$ is considered as a random variable (different from the classical mixed models where a design matrix is often considered as data), the expectation of the mixed model is E($y$) = $1\mu$ and the variance of the mixed mode is

$$\mathrm{var}(y) = \mathrm{var}(Z)\alpha^2 + A\sigma_\xi^2 + R\sigma^2 \tag{5}$$

This is an $n \times n$ variance matrix, where $n$ is the sample size. The total phenotypic variance and the partitioning of the total variance are shown below,

$$\sigma_P^2 = \frac{1}{n}\mathrm{tr}[\mathrm{var}(Z)]\alpha^2 + \frac{1}{n}\mathrm{tr}(A)\sigma_\xi^2 + \frac{1}{n}\mathrm{tr}(R)\sigma^2 \tag{6}$$

where $n^{-1}\mathrm{tr}[\mathrm{var}(Z)] = \sigma_z^2$, $n^{-1}\mathrm{tr}(A) = 1$ (assuming that no individuals are inbred), and $n^{-1}\mathrm{tr}(R)$ = 1 (assuming independent and homogeneous residual variance). Therefore,

$$\sigma_P^2 = \sigma_Z^2\alpha^2 + \sigma_\xi^2 + \sigma^2 \tag{7}$$

Let $\sigma_{QTL}^2 = \sigma_Z^2\alpha^2$ and the proportion of phenotypic variance contributed by the QTL is

$$h_{QTL}^2 = \frac{\sigma_{QTL}^2}{\sigma_P^2} = \frac{\sigma_Z^2\alpha^2}{\sigma_Z^2\alpha^2 + \sigma_\xi^2 + \sigma^2} \tag{8}$$

At this moment, we have dealt with the model and not mentioned any estimation of the QTL variance, which will be discussed later.

There is no doubt that model (1) or (2) is a mixed model because the same model includes both the fixed effects ($X\beta$) and the random effect ($Z\alpha + \xi$). However, in a typical mixed model, the design matrices ($X$ and $Z$) are treated as observed data and are considered as constants. In quantitative genetics, the design matrix $Z$ is considered as a variable and this makes the quantitative genetics model different from a typical linear mixed model. If $\alpha$ is considered as a fixed effect and $Z$ is considered as "data", the expectation of model (1) in a typical linear mixed model analysis would be E($y$) = $X\beta + Z\alpha$ and the variance matrix would be

$$\mathrm{var}(y) = A\sigma_\xi^2 + R\sigma^2 \tag{9}$$

Information about the QTL disappears from the variance, which was first notified by Gianola et al [21]. Therefore, model (9) is not a correct model for estimation of QTL variance. If we assign a normal distribution to $\alpha$ with mean zero and variance $\sigma_\alpha^2$, model (1) remains a mixed model with an expectation of E($y$) = $X\beta$ and a variance matrix of

$$\mathrm{var}(y) = ZZ^T\sigma_\alpha^2 + A\sigma_\xi^2 + R\sigma^2 \tag{10}$$

Now the QTL variance $\sigma_\alpha^2$ appears in the variance of $y$ and we can talk about QTL variance and the proportion of phenotypic variance contributed by the QTL. We now need to interpret $\alpha \sim N(0, \sigma_\alpha^2)$ for a single $\alpha$. According to Gianola et al [21], $N(0, \sigma_\alpha^2)$ is called a prior distribution

for $\alpha$ and $\sigma_\alpha^2$ is the prior variance or prior uncertainty. Since there is only one random draw from this distribution per population, the variance is defined as $\sigma_\alpha^2 = \alpha^2$.

## 2.2 Estimated QTL variance and QTL heritability

It is not surprising to see the following simple extension of Eq (8) to estimate the QTL heritability,

$$\hat{h}_{QTL}^{2(\text{BIASED})} = \frac{\sigma_Z^2 \hat{\alpha}^2}{\sigma_Z^2 \hat{\alpha}^2 + \hat{\sigma}_\xi^2 + \hat{\sigma}^2} \tag{11}$$

Unfortunately, this is not the correct estimate of QTL heritability (Luo et al. 2003) because $\hat{\sigma}_\alpha^2 \neq (\hat{\alpha})^2 = \hat{\alpha}^2$. The estimate is biased upward, especially when the sample size is small. The reason is that $\alpha$ is unknown and it is replaced by an estimate. However, the estimation is subject to an estimation error, $s_{\hat{\alpha}} = \sqrt{\text{var}(\hat{\alpha}|\alpha)}$, which has not played a role in Eq (11). The correct estimate of the QTL variance is

$$\hat{\sigma}_{QTL}^2 = \sigma_Z^2 \hat{\sigma}_\alpha^2 = \sigma_Z^2 (\hat{\alpha}^2 - s_{\hat{\alpha}}^2) \tag{12}$$

The estimated QTL heritability is simply

$$\hat{h}_{QTL}^2 = \frac{\sigma_Z^2 (\hat{\alpha}^2 - s_\alpha^2)}{\sigma_Z^2 (\hat{\alpha}^2 - s_\alpha^2) + \hat{\sigma}_\xi^2 + \hat{\sigma}^2} \tag{13}$$

It is a common practice to standardize the $Z$ variable prior to QTL mapping so that $Z = (Z^* - \mu_{Z^*})/\sigma_{Z^*}$, where $Z^*$ represents the $Z$ variable in its original scale, $\mu_{Z^*}$ and $\sigma_{Z^*}$ are the mean and standard deviation of the original $Z$ variable. The standardized $Z$ variable has E($Z$) = 0 and *var* ($Z$) = 1. Using the standardized $Z$ will result in

$$\hat{h}_{QTL}^2 = \frac{\hat{\alpha}^2 - s_{\hat{\alpha}}^2}{\hat{\alpha}^2 - s_{\hat{\alpha}}^2 + \hat{\sigma}_\xi^2 + \hat{\sigma}^2} \tag{14}$$

Hereafter, we use the standardized genotype indicator variable in all subsequent data analyses. Therefore, $\sigma_{QTL}^2 = \sigma_\alpha^2 = \alpha^2$.

## 2.3 Treating QTL effect as random

Terminologies like QTL variance and QTL heritability are defined in the context of a random QTL effect. However, all previous discussions are based on the fixed model framework for the QTL effect. Let us assume $\alpha \sim N(0, \sigma_\alpha^2)$. This treatment is a Bayesian analysis of QTL effect. Recall that the linear mixed model in (1) is,

$$y = X\beta + Z\alpha + \xi + \varepsilon \tag{15}$$

When the QTL effect is treated as random, the expectation of $y$ in Eq (15)) is E($y$) = $X\beta$ and the variance matrix of $y$ in Eq (15) is

$$\text{var}(y) = V = ZZ^T \sigma_\alpha^2 + A\sigma_\xi^2 + R\sigma^2 = (ZZ^T \lambda_\alpha + A\lambda_\xi + R)\sigma^2 = H\sigma^2 \tag{16}$$

where $\lambda_\alpha = \sigma_\alpha^2/\sigma^2$, $\lambda_\xi = \sigma_\xi^2/\sigma^2$ and $H = ZZ^T\lambda_\alpha + A\lambda_\xi + R$. The total phenotypic variance is partitioned below,

$$\sigma_P^2 = \frac{1}{n}\text{tr}(ZZ^T)\sigma_\alpha^2 + \frac{1}{n}\text{tr}(A)\sigma_\xi^2 + \frac{1}{n}\text{tr}(R)\sigma^2 = \sigma_\alpha^2 + \sigma_\xi^2 + \sigma^2 \tag{17}$$

where

$$\frac{1}{n}\mathrm{tr}(ZZ^T) = \frac{1}{n}\sum\nolimits_{j=1}^{n} Z_j^2 = \sigma_Z^2 = 1 \tag{18}$$

due to $Z$ being defined as a standardized variable ($E(Z) = 0$ and $\mathrm{var}(Z) = 1$).

We now introduce a restricted maximum likelihood (REML) method to estimate the QTL variance. Let $\theta = \{\sigma_\alpha^2, \sigma_\xi^2, \sigma^2\}$ be the three variance components. Given the expectation and the variance of model (15), the restricted log likelihood function is

$$L(\theta) = -\frac{1}{2}\ln|V| - \frac{1}{2}\ln|X^T V^{-1} X| - \frac{1}{2}(y - X\hat{\beta})^T V^{-1}(y - X\hat{\beta}) \tag{19}$$

where

$$\hat{\beta} = (X^T H^{-1} X)^{-1}(X^T H^{-1} y) \tag{20}$$

which is not a parameter but expressed as a function of $H$ and thus a function of $\theta$. Therefore, the likelihood function only contains three variance components, i.e., three parameters. Maximization of (19) with respect to $\theta$ yields the REML estimate of $\theta$, denoted by $\hat{\theta} = \{\hat{\sigma}_\alpha^2, \hat{\sigma}_\xi^2, \hat{\sigma}^2\}$. The price to pay for treating the QTL effect as random is that the solution is implicit and iterations are required for the REML estimate of the QTL variance. Given the estimated variance components, the estimated QTL heritability is

$$\hat{h}_{QTL}^2 = \frac{\hat{\sigma}_\alpha^2}{\hat{\sigma}_\alpha^2 + \hat{\sigma}_\xi^2 + \hat{\sigma}^2} \tag{21}$$

The variance matrix of the estimated $\theta$ can be obtained via the inverse of the information matrix,

$$\mathrm{var}(\hat{\theta}) = I^{-1}(\hat{\theta}) = \left[-\frac{\partial^2 L(\hat{\theta})}{\partial\theta\partial\theta^T}\right]^{-1} \tag{22}$$

The detail of $\mathrm{var}(\hat{\theta})$ is

$$\mathrm{var}(\hat{\theta}) = \mathrm{var}\begin{bmatrix} \hat{\sigma}_\alpha^2 \\ \hat{\sigma}_\xi^2 \\ \hat{\sigma}^2 \end{bmatrix} = \begin{bmatrix} \mathrm{var}(\hat{\sigma}_\alpha^2) & \mathrm{cov}(\hat{\sigma}_\alpha^2, \hat{\sigma}_\xi^2) & \mathrm{cov}(\hat{\sigma}_\alpha^2, \hat{\sigma}^2) \\ \mathrm{cov}(\hat{\sigma}_\alpha^2, \hat{\sigma}_\xi^2) & \mathrm{var}(\hat{\sigma}_\xi^2) & \mathrm{cov}(\hat{\sigma}_\xi^2, \hat{\sigma}^2) \\ \mathrm{cov}(\hat{\sigma}_\alpha^2, \hat{\sigma}^2) & \mathrm{cov}(\hat{\sigma}_\xi^2, \hat{\sigma}^2) & \mathrm{var}(\hat{\sigma}^2) \end{bmatrix} \tag{23}$$

The standard error of the estimated QTL heritability (ratio of variance components) can be approximated via the Delta method. Let

$$X = L_X^T\hat{\theta} = \begin{bmatrix} 1 & 0 & 0 \end{bmatrix}\begin{bmatrix} \hat{\sigma}_\alpha^2 \\ \hat{\sigma}_\xi^2 \\ \hat{\sigma}^2 \end{bmatrix} = \hat{\sigma}_\alpha^2 \tag{24}$$

and

$$Y = L_Y^T \hat{\theta} = \begin{bmatrix} 1 & 1 & 1 \end{bmatrix} \begin{bmatrix} \hat{\sigma}_\alpha^2 \\ \hat{\sigma}_\xi^2 \\ \hat{\sigma}^2 \end{bmatrix} = \hat{\sigma}_\alpha^2 + \hat{\sigma}_\xi^2 + \hat{\sigma}^2 \tag{25}$$

The variance-covariance matrix of $X$ and $Y$ is

$$\mathrm{var} \begin{bmatrix} X \\ Y \end{bmatrix} = \begin{bmatrix} \mathrm{var}\,(X) & \mathrm{cov}(X, Y) \\ \mathrm{cov}(X, Y) & \mathrm{var}\,(Y) \end{bmatrix} = \begin{bmatrix} L_X^T\,\mathrm{var}\,(\hat{\theta})L_X & L_X^T\,\mathrm{var}\,(\hat{\theta})L_Y \\ L_Y^T\,\mathrm{var}\,(\hat{\theta})L_X & L_Y^T\,\mathrm{var}\,(\hat{\theta})L_Y \end{bmatrix} \tag{26}$$

Let

$$\hat{h}_{QTL}^2 = \frac{\hat{\sigma}_\alpha^2}{\hat{\sigma}_\alpha^2 + \hat{\sigma}_\xi^2 + \hat{\sigma}^2} = \frac{X}{Y} \tag{27}$$

The approximate variance of the estimated QTL heritability via the Delta method is

$$\begin{aligned} var\,(\hat{h}_{QTL}^2) &\approx \left(\frac{X}{Y}\right)^2 \left[ \frac{\mathrm{var}\,(X)}{X^2} - 2\frac{\mathrm{cov}(X, Y)}{XY} + \frac{\mathrm{var}\,(Y)}{Y^2} \right] \\ &= \left(\frac{X}{Y}\right)^2 \left[ \frac{L_X^T\,\mathrm{var}\,(\hat{\theta})L_X}{X^2} - 2\frac{L_X^T\,\mathrm{var}\,(\hat{\theta})L_Y}{XY} + \frac{L_Y^T\,\mathrm{var}\,(\hat{\theta})L_Y}{Y^2} \right] \end{aligned} \tag{28}$$

The standard error of the estimated QTL heritability is

$$s_{\hat{h}_{QTL}^2} = \sqrt{\mathrm{var}\,(\hat{h}_{QTL}^2)} \tag{29}$$

## 2.4 The MM and REML estimates of QTL variance

When the genotype indicator variable is standardized, the estimated QTL variance presented in Eq (12) is rewritten as

$$\hat{\sigma}_\alpha^2 = \hat{\alpha}^2 - s_{\hat{\alpha}}^2 \tag{30}$$

This is a moment estimate of the QTL variance. Let us take the expectation of $\hat{\alpha}^2$,

$$\mathrm{E}(\hat{\alpha}^2) = \alpha^2 + s_{\hat{\alpha}}^2 = \sigma_\alpha^2 + s_{\hat{\alpha}}^2 \tag{31}$$

The moment method of estimation for $\sigma_\alpha^2$ is obtained by replacing $\mathrm{E}(\hat{\alpha}^2)$ by $\hat{\alpha}^2$ in Eq (31), which leads to

$$\hat{\alpha}^2 = \sigma_\alpha^2 + s_{\hat{\alpha}}^2 \tag{32}$$

We then solve for $\sigma_\alpha^2$ from Eq (32), resulting in an unbiased estimate of the QTL variance,

$$\hat{\sigma}_\alpha^2 = \hat{\alpha}^2 - s_{\hat{\alpha}}^2 \tag{33}$$

This method is called the moment method (MM). The MM estimate of QTL variance utilizes the result of a fixed model (the mixed model with the QTL effect being treated as a fixed effect). The REML method for estimation of QTL variance directly deals with a random model (the mixed model with the QTL effect being treated as a random effect). Estimates from the two different approaches are identical if negative estimates from MM are set to zero.

Let $\hat{\alpha}$ be the estimated QTL effect from the fixed model and $\text{var}(\hat{\alpha}|\alpha) = s_{\hat{\alpha}}^2$ be the squared estimation error. When the residual error of the fixed model in Eq (1) is normally distributed, the estimated QTL effect is also normally distributed, i.e., $\hat{\alpha} \sim N(\alpha, s_{\hat{\alpha}}^2)$. In this case, $\hat{\alpha}$ and $s_{\hat{\alpha}}^2$ are sufficient statistics of $\alpha$. To estimate the variance of $\alpha$ (the QTL variance $\sigma_\alpha^2$), we can simply obtain it from the sufficient statistics, not from the original data. Let us propose a random model for $\hat{\alpha}$ (treated as an observed data point),

$$\hat{\alpha} = \alpha + e_{\hat{\alpha}} \tag{34}$$

where $\alpha$ is the true value with a normal distribution $\alpha \sim N(0, \sigma_\alpha^2)$ and $e\hat{\alpha} \sim N(0, s_{\hat{\alpha}}^2)$ is the residual error with a known error variance. The expectation of model (34) is $\text{E}(\hat{\alpha}) = 0$ and the variance is

$$\text{var}(\hat{\alpha}) = \text{var}(\alpha) + \text{var}(e\hat{\alpha}) = \sigma_\alpha^2 + s_{\hat{\alpha}}^2 \tag{35}$$

The likelihood function from the sufficient statistics is

$$L(\sigma_\alpha^2) = -\frac{1}{2}\left[\ln(\sigma_\alpha^2 + s_{\hat{\alpha}}^2) + \frac{\hat{\alpha}^2}{\sigma_\alpha^2 + s_{\hat{\alpha}}^2}\right] \tag{36}$$

The ML solution is

$$\hat{\sigma}_\alpha^2 = \begin{cases} \hat{\alpha}^2 - s_{\hat{\alpha}}^2 & \text{for } \hat{\alpha}^2 > s_{\hat{\alpha}}^2 \\ 0 & \text{for } \hat{\alpha}^2 > s_{\hat{\alpha}}^2 \end{cases} \tag{37}$$

which is exactly the MM estimate of $\sigma_\alpha^2$ if negative solution is truncated at zero. A statistically more elegant notation for Eq (37) is

$$\hat{\sigma}_\alpha^2 = (\hat{\alpha}^2 - s_{\hat{\alpha}}^2)^+ \tag{38}$$

The equivalence between MM and REML will also be demonstrated empirically via Monte Carlo simulations later in the Result Section.

## 3 Data availability

### 3.1 Data of a working example from rice

Data and SAS codes used in the working example in the Result Section are given in Supplementary files. **S1 Data** contains the phenotypic values and the numerical codes (before and after standardization) of the genotypes for the locus of interest (Bin725), where the raw and standardized codes are named $z_0$ and $z$, respectively, and the phenotypic value is named $y$. **S2 Data** is the kinship matrix calculated from genome-wide markers (1619 bins). **S1 Code** contains the SAS codes of PROC MIXED for parameter estimation.

### 3.2 Data of an application to QTL mapping in mice

The mouse population consists of 110 $F_2$ mice derived from the cross between the B6 strain and the BTBR strain of mice [53]. The trait analyzed is the 10-week-body-weight. The mouse population was genotyped for 193 microsatellite markers over 19 autosomes with an average of 10 cM per marker interval. We added one pseudo marker in every 5 cM to generate a map with a total of 466 marker positions (193 real markers and 273 pseudo markers). An $n \times n = 110 \times 110$ kinship matrix was calculated from the 466 marker genotypes and this kinship matrix was used for QTL mapping under the polygenic model (**S3 Data**). The GLIMMIX procedure in SAS was used to analyze the data. PROC GLIMMIX is a very general procedure that

can handle generalized linear mixed models. The mouse data and the SAS code to analyze the data are provided in **S4 Data** and **S2 Code**, respectively.

# 4 Result

## 4.1 A working example

An IMF2 (immortalized $F_2$) population of rice with $n = 278$ hybrids was used as an example for illustration [54]. The trait is the 1000-grain weight (KGW). The experiment was replicated in two years (1998 and 1999). The average of the two- year replicates is the response variable for data analysis. There are $m = 1619$ bins (segregating markers) available for QTL mapping. The three genotypes (*A*, *H* and *B*) are coded as 1 for *A*, 0 for *H* and -1 for *B*. The kinship matrix was calculated from all 1619 markers. The kinship matrix was eventually normalized prior to the data analysis. A normalized kinship matrix has a property of $tr(K) = n$, i.e., the trace of *K* equals the sample size. The data were originally published by Hua et al. [54] and later by Xu et al. [55]. We illustrated the analysis of a single marker (Bin725) as an example. This locus is known to contain a QTL for grain width (GW) [56]. The numerically coded genotypes of this locus were standardized prior to the data analysis. The phenotypic values and the numerical codes (before and after standardization) of the genotypes for the locus (Bin725) are given in **S1 Data**. The raw and standardized codes are named $z_0$ and $z$, respectively. The phenotypic value is named *y*. The kinship matrix is provided in **S2 Data**. The SAS codes of PROC MIXED are given in **S1 Code**.

The same data were fitted to two models. One is the so called fixed model where the QTL effect was treated as a fixed effect. The other is the so called random model where the QTL effect was treated as a random effect. Both models have a random polygenic component and thus both are mixed models. **Table 1** shows the parameters estimated from the two models.

The fixed model estimates for the parameters are also the MM estimates while the random model estimates are called the REML estimates. The two methods are clearly the same (see **Table 1**). The MM method is computationally more robust than the REML method because it estimates two variance components while the REML method estimates three. The estimated QTL variance from the fixed model is

$$\hat{\sigma}^2_{QTL}(\text{fixed model}) = \hat{\alpha}^2 - \text{var}(\hat{\alpha}|\alpha) = 0.5278^2 - 0.1122^2 = 0.2660$$

and thus the estimated QTL heritability from the fixed model is

$$\hat{h}^2_{QTL}(\text{fixed model}) = \frac{\hat{\sigma}^2_{QTL}(\text{fixed})}{\hat{\sigma}^2_{QTL}(\text{fixed}) + \hat{\sigma}^2_{\xi} + \hat{\sigma}^2} + \frac{0.2660}{0.2660 + 3.5054 + 0.3842} = 0.06401$$

**Table 1. Estimated parameters of trait KGW from the fixed and random models for bin 725 of the rice population ($\sigma^2_{\alpha} = \alpha^2$).**

| Parameter | Fixed model | | Random model | |
|---|---|---|---|---|
| | Estimate | StdErr | Estimate | StdErr |
| $\alpha$ | 0.5278 | 0.1122 | 0.5039 | 0.1096 |
| $\sigma^2_{\alpha}$ | 0.2660 | - | 0.2660 | 0.3939 |
| $\sigma^2_{\xi}$ | 3.5054 | 0.5375 | 3.5038 | 0.5371 |
| $\sigma^2$ | 0.3842 | 0.0938 | 0.3845 | 0.0938 |

The estimated QTL heritability from the random model is

$$\hat{h}^2_{QTL}(\text{random model}) = \frac{\hat{\sigma}^2_{QTL}(\text{random})}{\hat{\sigma}^2_{QTL}(\text{random}) + \hat{\sigma}^2_\xi + \hat{\sigma}^2} == \frac{0.2660}{0.2660 + 3.5038 + 0.3845} = 0.06403$$

The two estimates are nearly identical. The standard error of the estimated QTL heritability can be obtained from the random model because we have an asymptotic variance-covariance matrix of the three estimated variance components (**Table 2**). Let

$$\hat{h}^2_{QTL} = \frac{X}{Y} = \frac{0.2660}{0.2660 + 3.5038 + 0.3845} = 0.06403$$

Define $L^T_X = \begin{bmatrix} 1 & 0 & 0 \end{bmatrix}$ and $L^T_Y = \begin{bmatrix} 1 & 1 & 1 \end{bmatrix}$, and let $\text{var}(\hat{\theta})$ be the $3 \times 3$ asymptotic variance matrix listed in **Table 2**. The delta approximation of the variance for the estimated QTL heritability is

$$\text{var}\left(\hat{h}^2_{QTL}\right) \approx \left(\frac{X}{Y}\right)^2 \left[\frac{L^T_X \text{var}(\hat{\theta})L_X}{X^2} - 2\frac{L^T_X \text{var}(\hat{\theta})L_Y}{XY} + \frac{L^T_Y \text{var}(\hat{\theta})L_Y}{Y^2}\right]$$

$$= \left(\frac{0.266}{4.1543}\right)^2 \times \left(\frac{0.1552}{0.266^2} - 2 \times \frac{0.155392}{0.266 \times 4.1543} + \frac{0.393749}{4.1543^2}\right)$$

$$= 0.007933$$

The standard error of the estimated QTL heritability is

$$s_{\hat{h}^2_{QTL}} = \sqrt{0.007933} = 0.089069$$

Note that the standard error is even larger than the estimated QTL heritability itself, due to the relatively small QTL variance and the small sample size ($n = 278$).

The naïve (biased) estimate of the QTL heritability is

$$\hat{h}^{2(\text{BIASED})}_{QTL} = \frac{\hat{\alpha}^2}{\hat{\alpha}^2 + \hat{\sigma}^2_\xi + \hat{\sigma}^2} + \frac{0.5278^2}{0.5278^2 + 3.5054 + 0.3842} = 0.066833$$

which is slightly higher than the unbiased estimate (0.06401). The relative bias is (0.06683–0.06401)/0.06401 = 4.4%.

We also investigated the $R^2$ of the mixed model with the QTL effect being treated as a fixed effect. Since the model is a mixed model with a random polygenic component, there is no easy way to calculate various sums of squares. As a result, we used the pseudo $R^2$ [50] to measure the model goodness of fit, which is an alternative way to measure the proportion of phenotypic variance contributed by a QTL. The likelihood ratio test statistic is

$$LRT = -2\left[L_0\left(\hat{\sigma}^2_\xi, \hat{\sigma}^2\right) - L_0\left(\hat{\alpha}, \hat{\sigma}^2_\xi, \hat{\sigma}^2\right)\right] = -2 \times (-464.4 + 455.05) = 18.7$$

**Table 2. Asymptotic variance-covariance matrix of the estimated variance components for trait KGW from the random model analysis (REML) for bin 725 of the rice population ($\sigma^2_\alpha = \alpha^2$).**

| var $(\hat{\theta})$ | $\sigma^2_\alpha$ | $\sigma^2_\xi$ | $\sigma^2$ |
|---|---|---|---|
| $\sigma^2_\alpha$ | 0.15520 | 0.00036 | -0.00020 |
| $\sigma^2_\xi$ | 0.00036 | 0.28840 | -0.02950 |
| $\sigma^2$ | -0.00020 | -0.02950 | 0.00881 |

The pseudo $R^2$ is

$$R^2_{\text{pseudo}} = 1 - \exp(-LRT/n) = 1 - \exp(-18.7/278) = 0.065054$$

which is higher than the unbiased $h^2_{QTL}$ (0.06401) and lower than the biased $h^2_{QTL}$ (0.06683).

## 4.2 Equivalence between the REML and MM estimates of a QTL variance

We fixed the population size at $n \times m = 10 \times 5 = 50$, where $n$ is the number of families and $m$ is the number of full siblings per family. The polygenic variance and the residual variance were fixed at $\sigma^2_\xi = \sigma^2 = 10$. A QTL was simulated with frequencies of $\Pr(A_1A_1) = 0.25$, $\Pr(A_1A_2) = 0.5$ and $\Pr(A_2A_2) = 0.25$, respectively. The numerical codes ($Z$ variable) for the three genotypes were set at 1, 0 and -1, respectively, for the three genotypes. The $Z$ variable was eventually standardized to have mean 0 and variance 1. Four simulation experiments were conducted under four different levels of QTL heritability ($h^2_{QTL}$): 0.05, 0.10, 0.15 and 0.20. The $\sigma^2_\alpha = \alpha^2$ values (true values) were calculated from

$$\alpha^2 = \left(\sigma^2_\xi + \sigma^2\right) \frac{h^2_{QTL}}{1 - h^2_{QTL}} = 20 \times \frac{h^2_{QTL}}{1 - h^2_{QTL}}$$

where $\sigma^2_\xi = 10$ and $\sigma^2 = 10$ are the polygenic and residual variances, respectively. The $\alpha^2$ values are 1.0526, 2.2222, 3.5294 and 5.0000, respectively, corresponding to the four different levels of QTL heritability. Each experiment was replicated 500 times. The estimated QTL variances from the fixed model (truncated moment) and the random model (REML) were compared by plotting the fixed model estimate against the random model estimate (**Fig 1**). All points of the scatter plots are on the diagonal lines except a couple of points slightly deviating from the diagonals. The simulations empirically validated that the truncated moment method is equivalent to the REML method. The slight deviations between the two methods is due to local convergence of the REML method because it involves three variance components while the truncated moment method involves only two variance components. In real data analysis, the random model analysis is not necessary because it is identical to the fixed model analysis and the latter is significantly faster than the former in terms of computational speed (see Discussion).

## 4.3 Bias of estimated QTL variance

**4.3.1 Single marker analysis.** This method is similar to interval mapping, where one QTL is included in a regression model and there is no polygenic background control for multiple QTL. To show the bias in estimated QTL variance and QTL heritability, we simulated data in the following scenarios. The residual variance was set at $\sigma^2 = 20$, the population mean was set at $\mu = 10$. The QTL heritability ranged from 0 to 0.2 incremented by 0.001. The QTL genotype indicator variable ($Z$) was generated from three genotypes with frequencies of 0.25, 0.5 and 0.25, respectively. The squared QTL effect corresponding to a given QTL heritability was calculated from

$$\alpha^2 = \frac{h^2_{QTL}}{1 - h^2_{QTL}} \sigma^2 = \frac{h^2_{QTL}}{1 - h^2_{QTL}} \times 20$$

The $Z$ variable was eventually standardized, i.e., $\mu_Z = 0$ and $\sigma^2_Z = 1$, prior to data analysis. The sample size varied at the following levels: 25, 50 100, 150, 200 and 250. A total of 500 replicated experiments were conducted under each scenario. The average of the 500 replicates was plotted. The estimated QTL variances (squared method, moment method and restricted maximum likelihood method) are plotted against the true QTL variance.

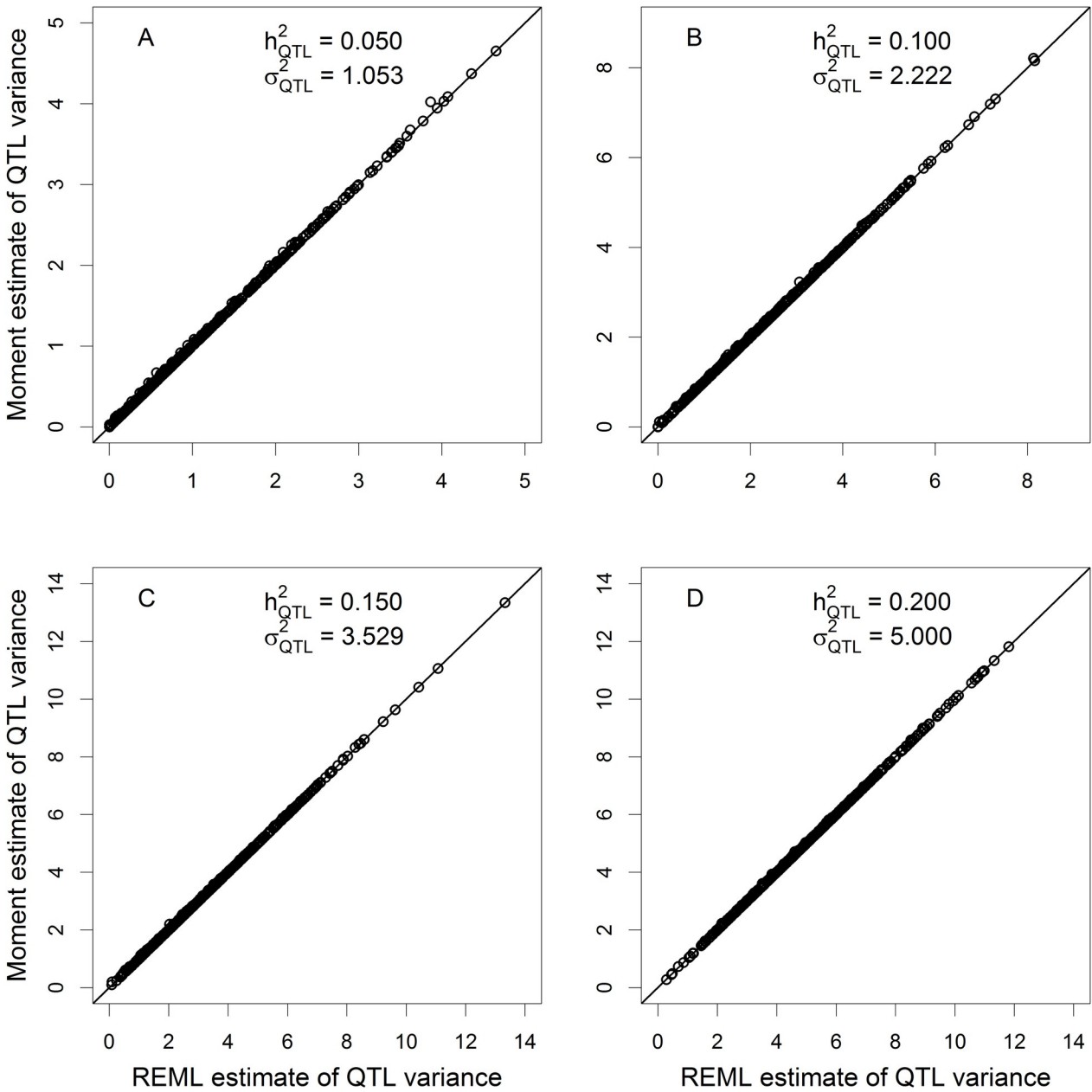

**Fig 1. Comparison of the estimated QTL variances from the fixed model and the random model.** Moment estimate (fixed model) of QTL variance plotted against REML estimate (random model) of QTL variance when the true QTL heritability is (A) 0.05, (B) 0.10, (C) 0.15 and (D) 0.20.

The results are shown in **Fig 2**. The naïve squared method (purple) is clearly biased upwards because the curves deviate far from the diagonal lines. The bias of the squared method is progressively reduced until the sample reaches 250 where the bias is barely noticeable. The REML method (blue curve) shows some bias when the true QTL variance is small and the sample is very small ($n = 25$ and $n = 50$), but the bias fades away quickly as the sample size reaches $n = 100$. The moment method (negative estimate is allowed) shows no bias in all sample sizes and in all range of the true QTL variance.

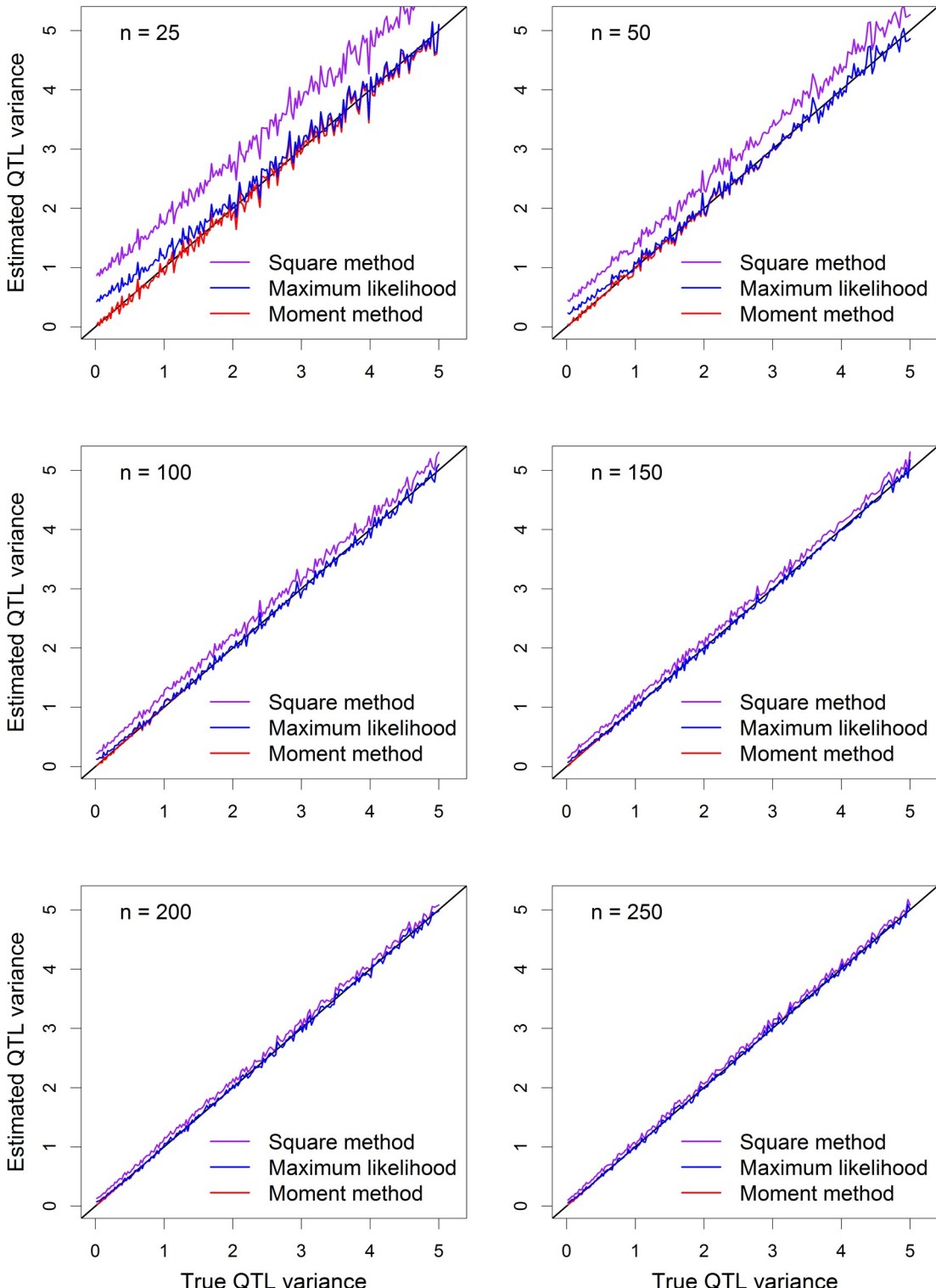

**Fig 2. Plots of estimated QTL variances from three methods against the true QTL variance.** The six panels of the figure show the results of six different sample sizes ($n$), which are 25, 50, 100, 150, 200 and 250, respectively.

We also compared the estimated QTL heritability under the six sample sizes. Here, we first calculated the average QTL variance estimated from 500 replicated simulation experiments. We then calculated the estimated QTL heritability from the average estimated QTL variance, as demonstrated below,

$$\hat{h}^2_{QTL} = \frac{\hat{\sigma}^2_{QTL}}{\hat{\sigma}^2_{QTL} + \hat{\sigma}^2}$$

The same trends observed for the estimated QTL variance were also observed here for the estimated QTL heritability (see **Fig 3**).

**4.3.2 Polygenic model analysis.** The model includes a polygenic effect to control the genetic background. Such a mixed model is the GWAS model [2] and the polygenic controlled QTL mapping procedure [16]. This model is analogous to the composite interval mapping where the genetic background is controlled by selected co-factors. We simulated full-sib family data with $m = 5$ full siblings per family. The number of families in a simulated population determines the sample size. We set the number of families at $n = 5,10,20,30,40,50$, corresponding to samples sizes $n \times m = 25,50,100,150,200,250$, respectively. The residual variance was set at $\sigma^2 = 10$ and the polygenic variance was set at $\sigma^2_\xi = 10$. The QTL heritability is defined as

$$h^2_{QTL} = \frac{\alpha^2}{\alpha^2 + \sigma^2_\xi + \sigma^2}$$

which allows us to calculate the QTL effect $\alpha$ via

$$\alpha^2 = \frac{h^2_{QTL}}{1 - h^2_{QTL}} \left( \sigma^2_\xi + \sigma^2 \right) = 20 \times \frac{h^2_{QTL}}{1 - h^2_{QTL}}$$

Again the genotypic code of a single QTL was standardized when used to generate and analyze the data. The true $h^2_{QTL}$ value ranges from 0 to 0.2 incremented by 0.001. Each experiment was replicated 500 times.

The plots of the average estimated QTL variance from the 500 replicated simulations against the true QTL variance are illustrated in **Fig 4**. The purple curves (the naïve squared method) is seriously biased in small samples ($n = 25$ and $n = 50$). However, the bias is very small for $n = 100$ and is barely noticeable when the sample size is above 150. The REML estimate is biased for $n = 25$ when the QTL variance is smaller than 2 (corresponding to QTL heritability of 0.075). The moment estimate of QTL variance is unbiased in all sample sizes and in the entire range of QTL variance. Similar trends were observed for the QTL heritability (see **Fig 5**).

Comparing **Fig 2** with **Fig 4** (also **Fig 3** with **Fig 5**), we realized that adding a polygene to the model can reduce the bias of the naïve squared method and the REML method relative to the corresponding single-marker analysis methods.

## 4.4 An application to QTL mapping for a mouse population

The mouse population consists of $n = 110$ $F_2$ mice genotyped for 193 markers. Adding 273 pseudo markers uniformly across the entire genome generated a map with an average of 5 cM per marker interval. The total number of marker positions is 193 + 273 = 466. We scanned the entire genome with two different models under two different strategies of QTL mapping. The two models are the fixed model and the random model. In the fixed model, the fixed effects included the intercept, the sex effect (1 for male and 0 for female) and the standardized marker genotype indicator variable. No random effect was included in the fixed model other than the

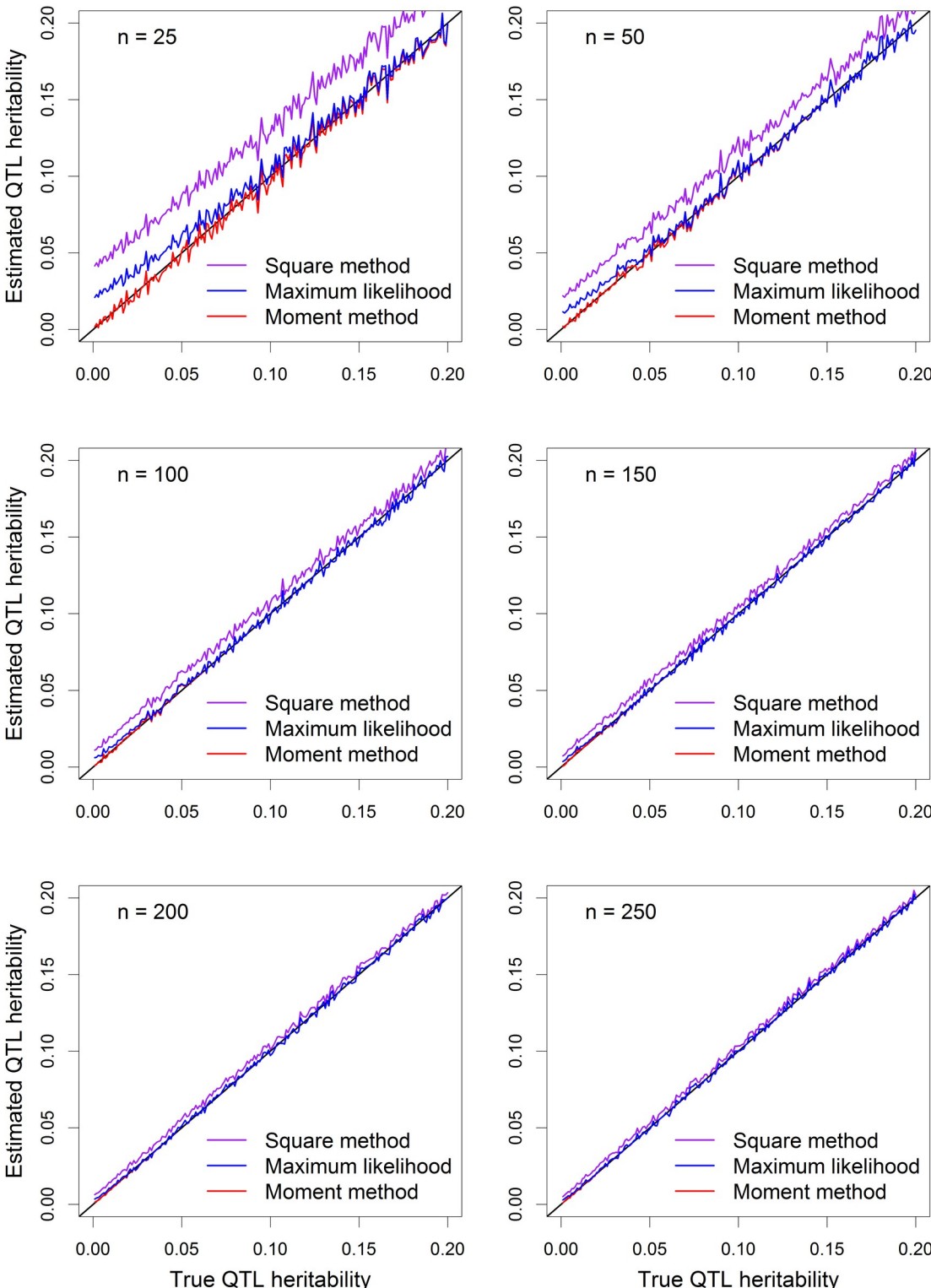

**Fig 3. Plots of estimated QTL heritability from three methods against the true QTL heritability.** The six panels of the figure show the results of six sample sizes (*n*), which are 25, 50, 100, 150, 200 and 250, respectively.

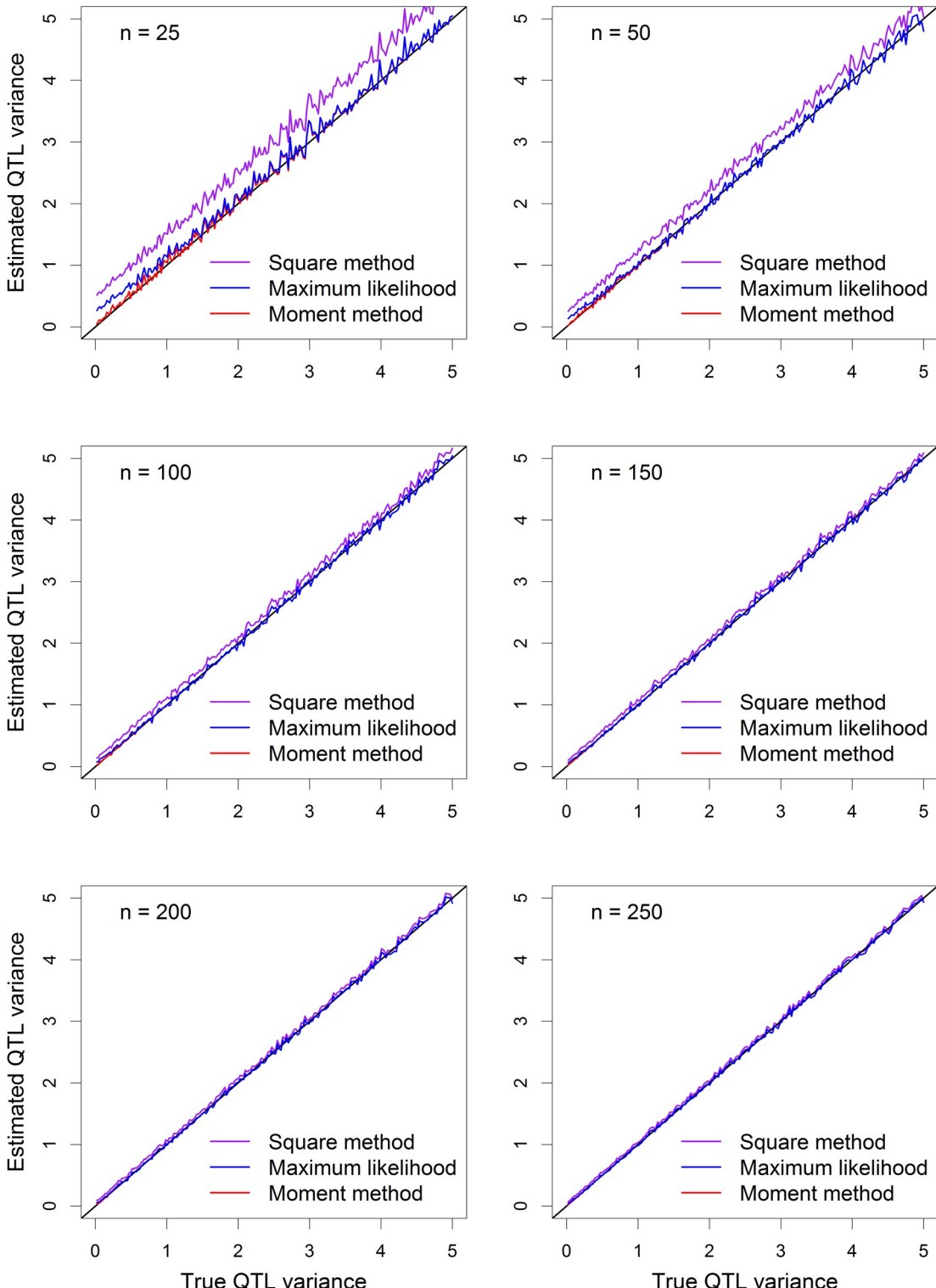

**Fig 4. Plots of estimated QTL variance against the true QTL variance under the polygenic model.** The six panels of the figure show the results of six sample sizes (*n*), which are 25, 50, 100, 150, 200 and 250, respectively.

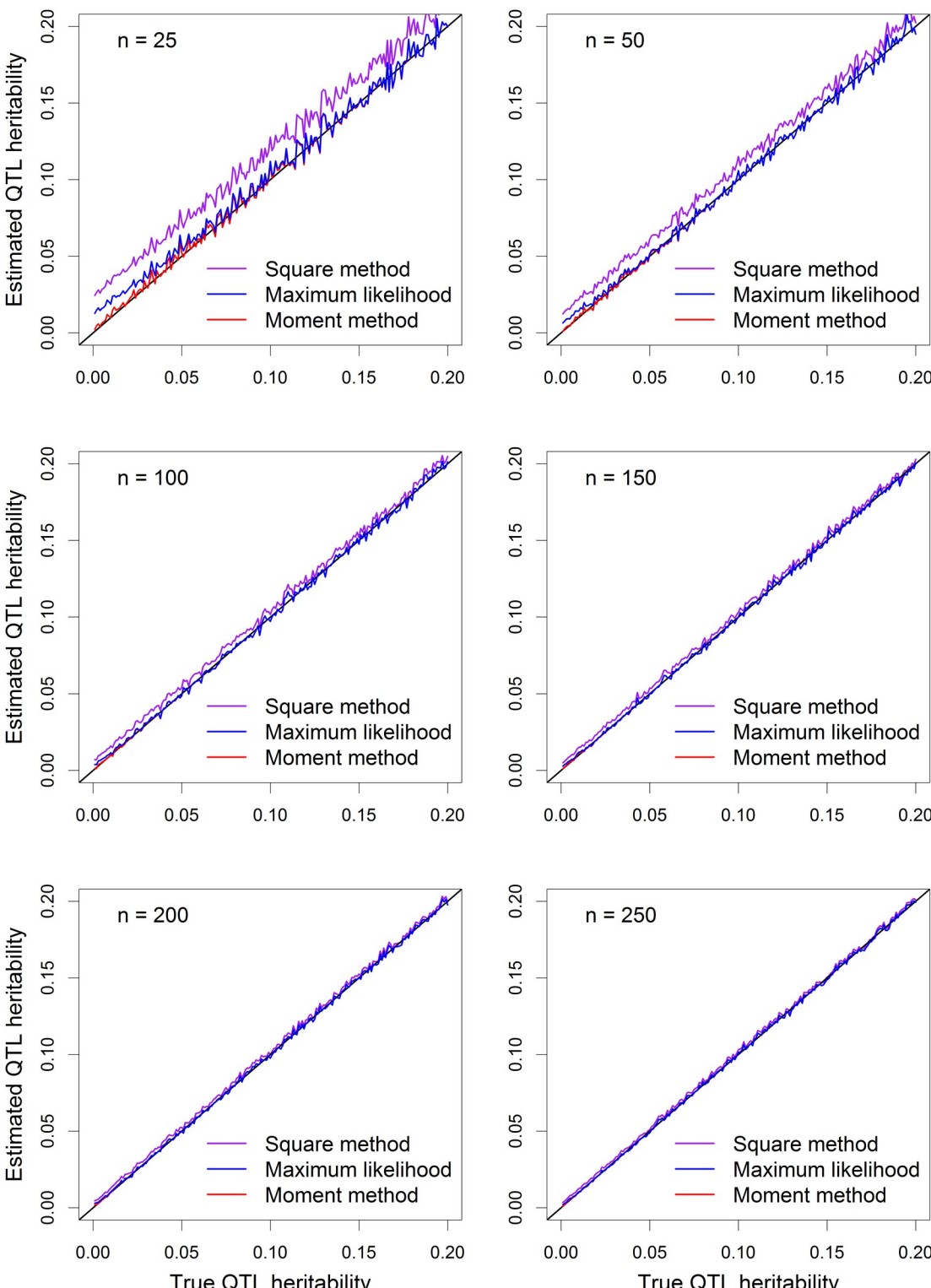

**Fig 5. Plots of estimated QTL heritability against the true QTL heritability under the polygenic model.** The six panels of the figure show the results of six sample sizes (*n*), which are 25, 50, 100, 150, 200 and 250, respectively.

residual error. For the random model, the standardized marker genotype indicator variable was included in the model as a random effect. The fixed effects included the intercept and the sex effect. The two QTL mapping strategies are the interval mapping procedure and the polygenic mapping procedure. The polygenic model, by definition, included a polygenic effect in the model to capture the polygenic background effect, while the interval mapping procedure does not include this polygenic effect.

Fig 6 shows the comparisons of the two models under the two strategies of QTL mapping for the 10-week-body-weight trait of the mouse population. The blue circles are the plots of the estimated QTL variances from the random model (QTL effect defined as a random effect) against the QTL variances from the fixed model moment method (QTL effect defined as a fixed effect). The red circles are the plots of the QTL variance from the squared effect method against the QTL variance from the fixed model moment method. Clearly, the random model and the fixed model moment methods are identical in the estimated QTL variance because the blue circles are all on the diagonals of the plots, while the QTL variance estimated from the squared effect method is consistently biased upward because the red circles are all above the diagonals of the plots. The left panels (Fig 6A and 6C) of the figure show the estimated QTL variances from the two models under the two QTL mapping procedures. The right panels (Fig 6B and 6D) of the figure compare the QTL heritability for the two models under the two QTL mapping procedures. The top panels (Fig 6A and 6B) of the figure show the result from the interval mapping procedure while the bottom panels (Fig 6C and 6D) show the result from the polygenic model analysis. Comparing the two QTL mapping procedures (interval mapping vs. polygenic mapping), the biases in estimated QTL variance (heritability) are greater for the polygenic method than the biases for the interval mapping procedure.

Fig 7 shows the Wald test statistic profiles and the QTL heritability profiles for the two QTL mapping procedures (interval mapping vs. polygenic mapping). The patterns of the profiles are much the same for the two procedures, but the profiles of the polygenic procedure have been substantially reduced compared to the interval mapping procedure. The threshold of the Wald test after Bonferroni correction is

$$t_W = \text{qchisq}(1 - 0.05/193, 1) = 13.34533 \tag{39}$$

where 193 is the number of real markers. The interval mapping procedure detected a significant marker on Chromosome 2 that is associated with the body weight trait of mice (Fig 7A and 7B). This marker, however, is not significant for the polygenic method (Fig 7C and 7D) due to strong shrinkage of the polygenic method. Interestingly, the marker with the highest Wald test statistic from the polygenic method is on Chromosome 18 with a Wald test statistic of 9.73 (not significant).

We now describe the marker on Chromosome 2 with the highest Wald test detected from the interval mapping procedure. This is a pseudo marker about 9 cM away from a real marker. The test statistic is $W = 17.72$ with $p = 0.00002559$. The estimated QTL heritability from the fixed model moment method, the random model method and the squared effect method are 0.1333, 0.1333 and 0.1401, respectively. The bias is (0.1401–0.1333)/0.1333 = 5.14%. None of the markers were significant from the polygenic method. The marker with the highest test statistic from the polygenic model analysis is on Chromosome 18 and it is a pseudo marker, 15 cM away from a real marker. The Wald test of this pseudo marker is $W = 9.73$ with a p-value of $p = 0.001812845$. The estimated QTL heritability are 0.1155, 0.1153 and 0.1271, respectively, for the fixed model moment method, the random model method and the squared effect method. The relative bias of the squared effect method is (0.1271–0.1153)/0.1153 = 10.21%.

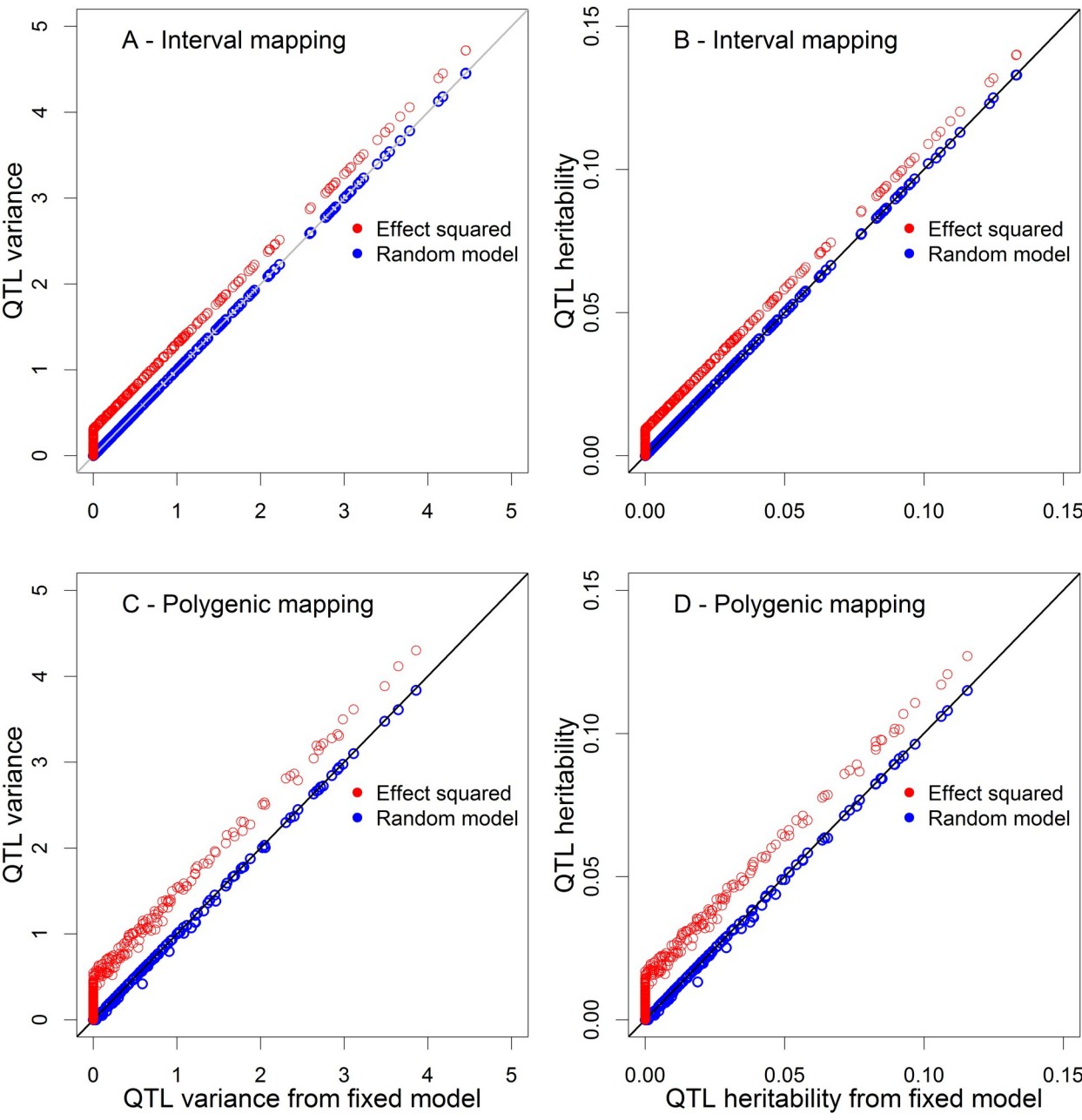

**Fig 6. Comparison of QTL variance and heritability from three estimation methods (squared, random model and fixed model).** (A) and (B) Plots of estimated QTL variance and heritability from the square method and the random model method against the estimates from the fixed model approach for interval mapping. (C) and (D) Plots of estimated QTL variance and heritability from the square method and the random model method against the estimates from the fixed model approach for polygenic mapping.

## 5 Discussion

In practice, the bias correction is only necessary for populations smaller than 200. Since most QTL mapping and GWAS experiments are conducted with sample sizes perhaps larger than 200, the current study is not intended to be read by crop and animal breeders. Tree breeders are a special group who often deal with small samples. QTL mapping and GWAS in trees may need bias correction for estimated QTL variances. The current study contributes more to the

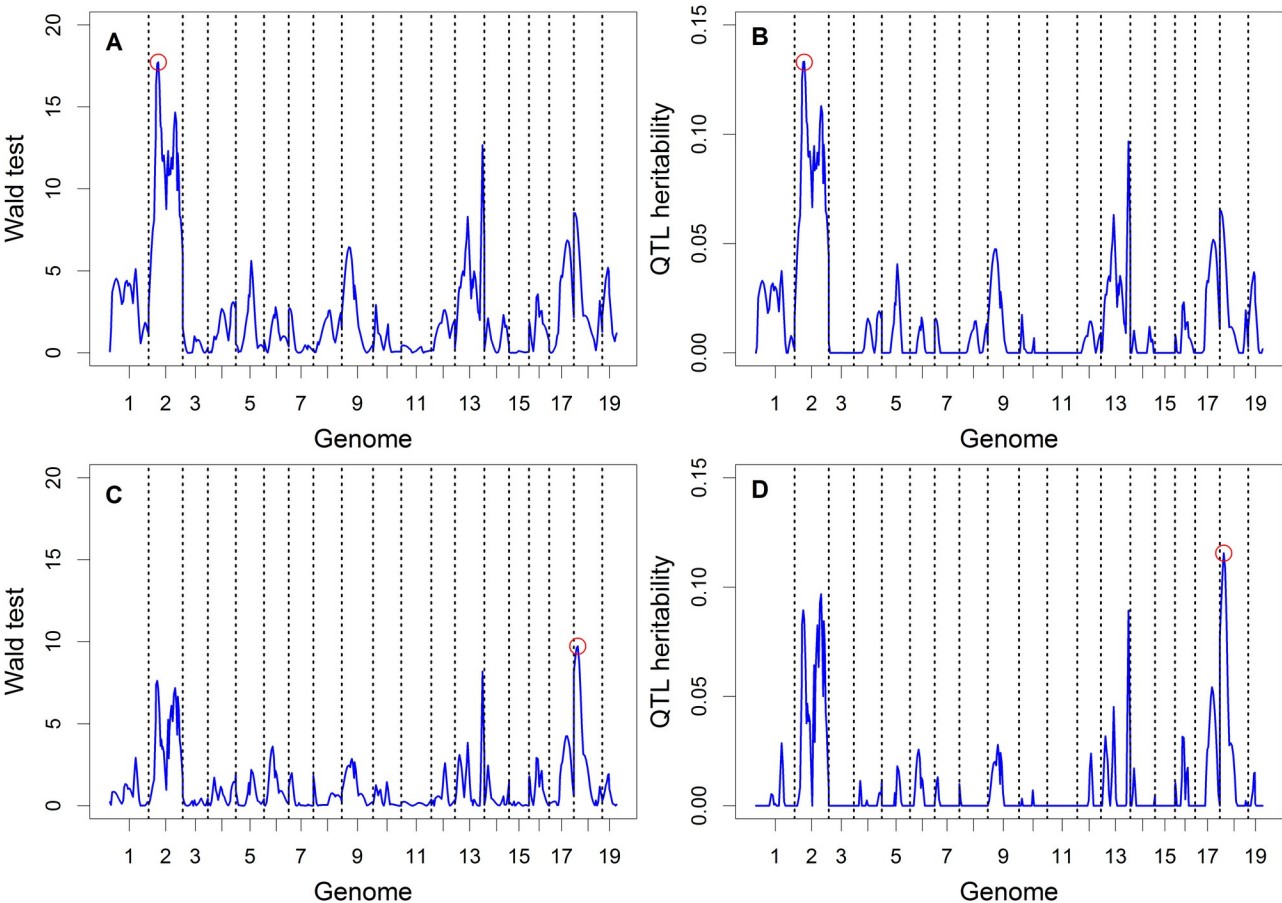

**Fig 7. Wald tests and estimated QTL heritability for body weight of the F$_2$ mouse population.** (A) and (C) show the Wald test statistics from interval mapping and polygenic mapping, respectively. (B) and (D) show the estimated QTL heritability from interval mapping and polygenic mapping, respectively.

quantitative genetics theory than to practical data analysis. Typical QTL mapping and GWAS models include QTL effects as fixed effects while the sizes of QTLs are reported as QTL variances or QTL heritability. The concept of variance does not go well with a fixed effect. It is the random effect that involves a variance. This conceptual relationship has been confused in the QTL mapping community for over three decades. This study has clarified this fundamental relationship.

Another fundamental contribution of this study to statistics is the "randomized fixed model approach" to estimating variances. If the number of levels of a random effect is small, this randomized fixed model can be used to estimate the variance associated with the random effect. We estimate the fixed effect as the best linear unbiased estimate (BLUE) and then convert the estimated fixed effect into a variance, like we presented in Eq (30) for the estimated QTL variance. A single regression coefficient is considered as just one level of a random effect. For multiple levels of a random effect, say $\alpha = [\alpha_1 \quad \alpha_2]^T$, and if each level of the random effect follows the same distribution, say $\alpha_k \sim N(0, \sigma_\alpha^2)$ for $k = 1,2$, the randomized fixed model approach to estimating $\sigma_\alpha^2$ is a simple extension of the MM estimate, as shown below,

$$\hat{\sigma}_\alpha^2 = \frac{1}{2}\text{tr}[\hat{\alpha}\hat{\alpha}^T - \text{var}(\hat{\alpha}|\alpha)] = \frac{1}{2}\sum\nolimits_{k=1}^{2}\left[\hat{\alpha}_k^2 - \text{var}(\hat{\alpha}_k|\alpha_k)\right] \tag{40}$$

where $\text{var}(\hat{\alpha}|\alpha)$ is a $2 \times 2$ variance matrix. In quantitative genetics, the four epistatic effects per pair of loci (additive × additive, additive × dominance, dominance × additive and dominance × dominance) may be modeled as a random effect with four group levels. Each level of the epistatic effects follows the same normal distribution denoted by $N(0, \sigma_{\text{EP}}^2)$. The epistatic variance ($\sigma_{\text{EP}}^2$) may be easily estimated using Eq (40) with the 2 levels in the formula substituted by 4 levels of the epistatic model. The original random (or mixed) model methodology does not have an explicit estimate of a variance component for the ML and REML methods. The randomized fixed model provides an explicit solution. As a result, this new method is much like the Type3 method of the MIXED procedure in SAS, but it is the QTL mapping version of the Type3 method.

Variance components may also be estimated via the Bayesian method by assigning a prior distribution to each variance components. The Bayesian estimate of $\sigma_\alpha^2 = \alpha^2$ is drastically different from the maximum likelihood estimate of $\sigma_\alpha^2$ in the situation of regression analysis. The reason is that the variance is defined and estimated from a "single group level." A good Bayesian estimate of a variance component needs at least three group levels [57–59]. The MCMC procedure in SAS was used to implement the Bayesian method for parameter estimation. Since coding PROC MCMC for the polygenic model using the marker-inferred kinship matrix is very difficult, we only investigated the simple model without the polygenic background control. Five different prior distributions were investigated, including the uniform prior on $\sigma_\alpha^2$ and a weakly informative half-Cauchy prior on $\sigma_\alpha$. Please see **S1 Note** for a complete list of prior distributions investigated in this project. **Table A in S1 Note** shows the results of Bayesian estimates of parameters in comparison with the estimates of the restricted maximum likelihood methods for the hybrid rice data (data of the working example). The estimated QTL effects and residual variances across a range of prior distributions are much the same compared with the estimates from the restricted maximum likelihood methods. However, the Bayesian estimates of the QTL variance are drastically different from the REML estimate and the differences are highly dependent of the prior distributions. Therefore, the proposed MM and REML estimate of the QTL variance under one group level may be the only option for estimating QTL heritability.

An alternative method to estimate the QTL heritability is the R squared ($R^2$), which does not rely on an estimated QTL variance. It requires partitioning of the total sum of squares into the regression sum of squares and the residual sum of squares. The ratio of the regression sum of squares to the total sum of squares is the R squared. This R squared is also called the coefficient of model determination, the model goodness of fit and so on. Take the interval mapping (single marker analysis without control for the polygenic background) for example, the regression model is

$$y = \mu + Z\alpha + \varepsilon \tag{41}$$

where $\mu$ is the intercept and $\sigma_Z^2 = 1$ because variable $Z$ has been standardized. The naïve estimate of the QTL heritability is defined as

$$\hat{h}_{QTL}^{2(\text{BIASED})} = \frac{\hat{\alpha}^2}{\hat{\alpha}^2 + \hat{\sigma}^2} \tag{42}$$

The R squared is defined as

$$R^2 = \frac{SS_{REG}}{SS_{REG} + SS_{RES}}$$

where

$$SS_{REG} = \hat{\alpha}^2 \sum_{j=1}^{n} (Z_j - \bar{Z})^2 = (n-1)\sigma_Z^2\hat{\alpha}^2 = (n-1)\hat{\alpha}^2$$

is the regression sum of squares and

$$SS_{RES} = \sum_{j=1}^{n} (y_j - \hat{\mu} - Z_j\hat{\alpha})^2 = (n-2)\hat{\sigma}^2$$

is the residual sum of squares. The R squared is

$$R^2 = \frac{(n-1)\hat{\alpha}^2}{(n-1)\hat{\alpha}^2 + (n-2)\hat{\sigma}^2} = \frac{\hat{\alpha}^2}{\hat{\alpha}^2 + \hat{\sigma}^2(n-2)/(n-1)} \tag{43}$$

Comparing Eq (43) with Eq (42), we conclude that $R^2 > \hat{h}^2_{QTL}$. The equality is only approached asymptotically. Since $\hat{h}^{2(BIASED)}_{QTL}$ defined in Eq (42) is already biased, the R squared is certainly biased as well.

The adjusted R squared, however, is a modification of the original R squared by taking into account the model size (number of independent variables). After a few steps of manipulation, the adjusted R squared can be expressed as

$$R^2_{ADJ} = 1 - (1-R^2) \times \frac{n-1}{n-2} = \frac{\hat{\alpha}^2 - \hat{\sigma}^2/(n-1)}{\hat{\alpha}^2 - \hat{\sigma}^2/(n-1) + \hat{\sigma}^2(n-2)/(n-1) + \hat{\sigma}^2/(n-1)} \tag{44}$$

We now re-write the estimated QTL heritability as

$$\hat{h}^2_{QTL} = \frac{\hat{\alpha}^2 - s_{\hat{\alpha}}^2}{\hat{\alpha}^2 - s_{\hat{\alpha}}^2 + \hat{\sigma}^2} = \frac{\hat{\alpha}^2 - \hat{\sigma}^2/(n-1)}{\hat{\alpha}^2 - \hat{\sigma}^2/(n-1) + \hat{\sigma}^2(n-2)/(n-1) + \hat{\sigma}^2/(n-1)} \tag{45}$$

where $s_{\hat{\alpha}}^2 = \hat{\sigma}^2/[(n-1)\sigma_Z^2] = \hat{\sigma}^2/(n-1)$ and $\hat{\sigma}^2(n-2)/(n-1) + \hat{\sigma}^2/(n-1) = \hat{\sigma}^2$ due to the fact that $\sigma_Z^2 = 1$. Eqs (44) and (45) are identical and thus $\hat{h}^2_{QTL} = R^2_{ADJ}$. The bias corrected heritability is not the R squared goodness of fit but it is identical to the adjusted R squared.

The additive genetic variance of a quantitative trait locus presented in Eq (4) is given in classical quantitative genetics textbooks [19,20]. Surprisingly, it is the result of a fixed model treatment of the QTL effect. The naïve estimate of the QTL variance converted from the squared effect is biased. With the random model, we can directly estimate the QTL variance $\sigma_\alpha^2 = \alpha^2$ via the REML method. However, there is no explicit solution for the REML estimation of the QTL variance, even if the model is a single marker model without polygene. The MM method is derived under the fixed model, but the MM estimate is identical to the REML estimate. Under the single marker model (without the polygene), the MM estimate is explicit, much more convenient to achieve than the random model REML estimate. Under the polygenic model with the QTL effect being considered as a fixed effect, there are two variance components, $\sigma_\xi^2$ and $\sigma^2$, explicit solution of the variance parameters are not available anyway. The fixed model approach ($\theta_{\text{Fixed}} = \{\sigma_\xi^2, \sigma^2\}$) to estimating the QTL variance has two advantages over the random model approach ($\theta_{\text{Random}} = \{\sigma_\alpha^2, \sigma_\xi^2, \sigma^2\}$). (1) In variance component analysis, adding one more variance can substantially increase the computational time, especially in large samples. Furthermore, in QTL mapping and GWAS, we are talking about estimating one more variance component for every locus while the total number of marker loci can be up to millions. (2) Adding one more variance component to the parameter array can complicate the landscape of the restricted log likelihood function and increase the risk of local convergence.

Regarding the computational times, we compared the fixed model and the random model of PROC MIXED in SAS for the working example (IMF2 rice population with 278 lines at bin725), the fixed model and random model took 3.40 and 3.49 CPU seconds, respectively. The corresponding numbers of iterations required for convergence were 6 and 5 for the fixed and random models, respectively. We then scanned the entire genome of 1619 markers for the same rice population with both the fixed model and the random model of PROC MIXED in SAS. The fixed model took 1 hour, 31 minutes and 38 seconds of CPU time to complete the scanning. The random model, however, took 2 hours, 20 minutes and 39 seconds of CPU time. The average numbers of iterations (over the 1619 loci) required for convergence were 11.73 and 12.25 for the fixed model and the random model, respectively.

Our common belief in QTL mapping and GWAS is that there are just a few detectable QTL per experiment. When the sample size is very large, more QTL can be detected. The current GWAS in human height has identified 83 associated SNPs with a sample size as large as 711428 individuals [60]. We are not interested in presenting QTL variances or QTL heritability of the entire genome; rather, only variances of detected QTL need to be presented. Therefore, we can still use the fixed effect model to scan the genome and only go back to the significant loci to calculate the QTL variances. In this case, we can tolerate the extra cost of REML estimation of QTL variances for the limited number of significant loci. One advantage of the REML estimation for the QTL variance is that an asymptotic variance matrix for $\hat{\theta}_{\text{Random}} = \left\{\hat{\sigma}_\alpha^2, \hat{\sigma}_\xi^2, \hat{\sigma}^2\right\}$ is available in PROC MIXED. This matrix allows us to calculate an approximate standard error of the estimate QTL heritability via the Delta approximation. Under the fixed model, however, we only have the asymptotic variance matrix for $\hat{\theta}_{\text{Fixed}} = \left\{\hat{\sigma}_\xi^2, \hat{\sigma}^2\right\}$; the variance for $\hat{\sigma}_\alpha^2 = (\hat{\alpha} - s_{\hat{\alpha}}^2)^+$ and the covariance between $\hat{\sigma}_\alpha^2$ and $\hat{\theta}_{\text{Fixed}}$ are not available. Therefore, we cannot calculate the standard error of the estimated QTL heritability. Another advantage of the random model (treating QTL effects as random) is to calculate the heritability of a QTL in a multiple QTL model when linkage disequilibrium (LD) among markers is present. This problem has been investigated by Gianola et al. [23] under the fixed effect model. We now review the fixed model approach using three loci as an example. The linear mixed model is

$$y = X\beta + \underbrace{Z_1\alpha_1 + Z_2\alpha_2 + Z_3\alpha_3}_{g} + \xi + \varepsilon \tag{46}$$

where $Z_k$ is the genotype indicator variable for QTL $k$ and $\alpha_k$ is the effect of QTL $k$ for $k = 1,2,3$. If these QTL are treated as fixed effects, the genetic variance associated with the three loci is

$$\text{var}(g) = \sigma_{Z_1}^2\alpha_1^2 + \sigma_{Z_2}^2\alpha_2^2 + \sigma_{Z_3}^2\alpha_3^2 + 2\sigma_{Z_1 Z_2}\alpha_1\alpha_2 + 2\sigma_{Z_1 Z_3}\alpha_1\alpha_3 + 2\sigma_{Z_2 Z_3}\alpha_2\alpha_3 \tag{47}$$

where $\sigma_{Z_k}^2$ is the variance of $Z_k$ and $\sigma_{Z_k Z_{k'}}$ is the covariance between $Z_k$ and $Z_{k'}$. This covariance is called the linkage disequilibrium (LD). The variance contributed by the $k$th locus is [23]

$$\sigma_{\alpha_k}^2 = \sigma_{Z_k}^2\alpha_k^2 + \sum_{k' \neq k}^{3}\sigma_{Z_k Z_{k'}}\alpha_k\alpha_{k'} \tag{48}$$

This QTL variance looks very strange because the variance of QTL $k$ contains effects of other QTL. The genetic variance contributed by the three loci is collectively expressed by

$$\text{var}(g) = \sum_{k=1}^{3}\sigma_{\alpha_k}^2 = \sigma_{\alpha_1}^2 + \sigma_{\alpha_2}^2 + \sigma_{\alpha_3}^2 \tag{49}$$

When the LD is absent, the covariance $\sigma_{Z_k Z_{f'}}$ disappears and the variance of QTL $k$ does not

contain effects of other QTL. If the effects of all QTL are treated as random effects, the genetic variance contributed by the three loci is

$$\sigma_g^2 = \frac{1}{n}\mathrm{tr}\left(Z_1 Z_1^T\right)\sigma_{\alpha_1}^2 + \frac{1}{n}\mathrm{tr}\left(Z_2 Z_2^T\right)\sigma_{\alpha_2}^2 + \frac{1}{n}\mathrm{tr}\left(Z_3 Z_3^T\right)\sigma_{\alpha_3}^2 \tag{50}$$

where $\sigma_{\alpha_k}^2 = \alpha_k^2$ for $k$ = 1,2,3. If the genotype indicator variables are standardized, $n^{-1}\mathrm{tr}\left(Z_k Z_k^T\right) = 1$ for all $k$ = 1,2,3, the above genetic variance is simplified into

$$\sigma_g^2 = \sigma_{\alpha_1}^2 + \sigma_{\alpha_2}^2 + \sigma_{\alpha_3}^2 = \alpha_1^2 + \alpha_2^2 + \alpha_3^2 \tag{51}$$

Therefore, treating multiple QTL effects as random has substantially simplified the genetic variance contributed by each QTL, even if LD is present.

A statistically significant QTL does not mean that it is significant biologically if the QTL contributes a very small proportion of the trait variance. However, a large QTL of biologically significance may not be significant statistically. The lack of power to detect such a QTL is primarily due to interactions of the QTL with other QTLs in linkage disequilibrium. For example, if two QTLs are in high LD but one is an antagonist of the other, neither one may be detected because the effect of one locus is cancelled by the other. A multiple locus model may improve the detection of both loci. More importantly, each locus may act as a member in a genetic network that consists of many loci [61]. An individual locus may not be detected alone but is detectable collectively as a network.

Regarding to the extra computational cost for correcting the bias, if the fixed model is used, there is no extra cost, because the correction only needs the intermediate results of QTL mapping and GWAS, i.e., the estimated QTL effect and the squared estimation error per locus. The intermediate results are often provided in the output files of QTL mapping and GWAS software packages. It appears that the correction method requires the genotype indicator variable ($Z$) to be standardized prior to the data analysis. In fact, this assumption is presented for ease of presentation and is not absolutely required. If the results of QTL mapping and GWAS are obtained from an unstandardized $Z$ variable, we simply calculate the QTL variance using $\hat{\sigma}_{QTL}^2 = \hat{\sigma}_Z^2[\hat{\alpha}^2 - \mathrm{var}(\hat{\alpha}|\alpha)]$, not $\hat{\sigma}_{QTL}^2 = \hat{\alpha}^2 - \mathrm{var}(\hat{\alpha})$. Calculation of $\hat{\sigma}_Z^2$ for each locus presents some extra computational burden.

## Supporting information

**S1 Data. Genotype indicator variables and phenotypic values of 1000 grain weight (KGW) of rice from 278 hybrids.** Column 1 (hybrid): IDs of 278 hybrid rice; Column2 (bin): this the bin ID (Bin 725) chosen from a total of 1619 bins. Column 3 (y): This is the average phenotypic value of KGW collected from 1998 and 1999. Column 4 (z0): This is the genotype indicator variable, 1, 0, -1, representing the three genotypes, A, H and B, respectively. Column 5 (z): this is the standardized genotypic indicator variable, z = (z0 −mean(z0))/stdev(z0).
(XLSX)

**S2 Data. Marker inferred kinship matrix among the 278 hybrid rice.** The matrix has been normalized so that all diagonal elements are unity. The first column (parm) holds a value of 1 and the second column (row) holds the row number of the kinship matrix. col1 −col278 represent the column names of the kinship matrix. This format is required by PROC MIXED (a SAS procedure).
(XLSX)

**S3 Data. Marker inferred kinship matrix among the 110 mice.** The matrix has been normalized so that all diagonal elements are unity. The first column (parm) holds a value of 1 and the

second column (row) holds the row number of the kinship matrix. col1 –col110 represent the column names of the kinship matrix. This format is required by PROC MIXED and PROC GLIMMIX (SAS procedure).
(CSV)

**S4 Data. This file contains the genotype indicator variable (z0) and the standardized version of this variable (z).** It also contains the phenotypic values of the 10-week-body-weight trait for all the 110 mice (y). The x variable indicates the sex, 1 for male and 0 for female. The data are sorted by markers.
(CSV)

**S1 Code. SAS code to read the data and estimate the variance components from the fixed model and the random model of PROC MIXED.** The first block of codes calls PROC MIXED with the QTL effect being treated as a random effect. The second block of codes calls PROC MIXED with the QTL effect being treated as a fixed effect.
(SAS)

**S2 Code. SAS code to read the data and perform QTL mapping for the mouse population.** The file contains four blocks of SAS program, one for each of the four combinations of the two models (fixed and random models) and the two procedures (interval and polygenic mappings). PROC GLIMMIX was used to perform the QTL mapping.
(SAS)

**S1 Note. Description of the Bayesian method and the Bayesian estimate of QTL variance ($\sigma_\alpha^2$). Table A:** Estimated parameters for trait KGW from the fixed model, the random model and the Bayesian analysis for bin 725 of the IMF2 rice population.
(DOCX)

## Author Contributions

**Conceptualization:** Shizhong Xu.

**Data curation:** Shizhong Xu.

**Formal analysis:** Shibo Wang, Fangjie Xie, Shizhong Xu.

**Funding acquisition:** Shizhong Xu.

**Investigation:** Shibo Wang, Fangjie Xie, Shizhong Xu.

**Methodology:** Shibo Wang, Fangjie Xie, Shizhong Xu.

**Project administration:** Shizhong Xu.

**Resources:** Shizhong Xu.

**Software:** Shizhong Xu.

**Supervision:** Shizhong Xu.

**Validation:** Shizhong Xu.

**Visualization:** Shizhong Xu.

**Writing – original draft:** Shibo Wang, Fangjie Xie, Shizhong Xu.

**Writing – review & editing:** Shibo Wang, Fangjie Xie, Shizhong Xu.

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
