## [Decision Letter · Decision Letter 0]

17 Nov 2021

Dear Prof. Xu,

Thank you very much for submitting your manuscript "Estimating Genetic Variance Contributed by a Quantitative Trait Locus: A Random Model Approach" for consideration at PLOS Computational Biology.

As with all papers reviewed by the journal, your manuscript was reviewed by members of the editorial board and by several independent reviewers. In light of the reviews (below this email), we would like to invite the resubmission of a significantly-revised version that takes into account the reviewers' comments.

We cannot make any decision about publication until we have seen the revised manuscript and your response to the reviewers' comments. Your revised manuscript is also likely to be sent to reviewers for further evaluation.

Sincerely,

Mingyao Li

Associate Editor

PLOS Computational Biology

Jian Ma

Deputy Editor

PLOS Computational Biology

Reviewer's Responses to Questions

**Comments to the Authors:**

Reviewer #1: This manuscript proposes a random-effect model approach to estimating the QTL variance.

The method reformulates the QTL model by treating the QTL effect as random and directly estimate the QTL variance (as a variance component) or adjust the bias by taking into account the error of the estimated QTL effect. A moment method of estimation has been proposed to correct the bias. The method has been validated via Monte Carlo simulation studies. The method has been applied to QTL mapping for the 10-week-body-weight trait from an F2 mouse population. The manuscript was well written, and developed a novel method that can be applied to many real data sets. I evaluate the work as a useful contribution and can be published. I have two comments.

1. The manuscript includes too many equations and derivations, some of which are easily derived and standard. I recommend simplify the mathematical presentation and thus improve readable.

2. The proposed random-effect assumes that the QTL effects follow a normal prior. What is the prior on the prior variance? You use an uniform prior. However, it results in a estimation towards zero. Gelman et al. Bayesian Data Analysis (Chapter 5) suggests weakly informative prior, which can solve the problem.

Reviewer #2: This manuscript presents an old but important question in QTL mapping and GWAS; i.e., how to correct biased genetic variance explained by a single QTL. The manuscript was well written and results are scientifically sound. I only have a minor comment.

In this reviewer's opinion, mapping and estimating a single QTL is not meaningful in practical breeding schemes. As claimed by the authors, a small-effect but statistically significant QTL is not useful in practice. Yet, such QTLs are very commonly detected in plant, animal and human genetic studies. According to a recent study, an insignificant locus by statistical testing is not necessarily insignificant on its merit, rather its effect is compromised by negative regulators (Wang et al. 2021). I believe that an in-depth discussion on this issue is crucial for strengthening this manuscript's quality and impact.

Wang HJ, et al. (2021) Modeling genome-wide by environment interactions through omnigenic interactome networks. Cell Reports 35: 109114.

Reviewer #3: The paper tries to address bias in h2 estimate due to non Beavis effect theoretically and empirically. However, their results suggest that such bias is negligible as long as n is not too small. Given that Beavis effects over dominant non Beavis ones, practically, it is thus important to simultaneously correct both for studies with very small n, which should be properly addressed. Otherwise, this research has very limited practical usages. Some other concerns

1. Not agree with the comments made between lines 277-282. The variance in (10) is a conditional variance of y given X and Z. The conditional variance of y given X only will depend on Z, so does the marginal variance of y.

2. Equations (16)-(18) are very confusing. So are equations (18) and (19), and it is not clear what is the difference between the two hat estimates of alpha in these two equations.

3. Section 2 is very wordy and can be significantly shortened. The authors should only present the models and derive the biases concisely without lengthy explanations.

**Have the authors made all data and (if applicable) computational code underlying the findings in their manuscript fully available?**

Reviewer #1: None

Reviewer #2: Yes

Reviewer #3: Yes

PLOS authors have the option to publish the peer review history of their article (what does this mean?). If published, this will include your full peer review and any attached files.

Reviewer #1: No

Reviewer #2: No

Reviewer #3: No
---

## [Decision Letter · Decision Letter 1]

13 Feb 2022

Dear Prof. Xu,

We are pleased to inform you that your manuscript 'Estimating Genetic Variance Contributed by a Quantitative Trait Locus: A Random Model Approach' has been provisionally accepted for publication in PLOS Computational Biology.

Best regards,

Mingyao Li

Associate Editor

PLOS Computational Biology

Jian Ma

Deputy Editor

PLOS Computational Biology

Reviewer's Responses to Questions

**Comments to the Authors:**

Reviewer #1: The authors have nicely addressed my previous comments. The revised manuscript has been improved. I have no further concerns.

Reviewer #2: The authors have satisfactorily addressed my concern.

**Have the authors made all data and (if applicable) computational code underlying the findings in their manuscript fully available?**

Reviewer #1: None

Reviewer #2: None

PLOS authors have the option to publish the peer review history of their article (what does this mean?). If published, this will include your full peer review and any attached files.

Reviewer #1: No

Reviewer #2: No

---

## [Editor Report · Acceptance letter]

4 Mar 2022

PCOMPBIOL-D-21-01711R1 

Estimating Genetic Variance Contributed by a Quantitative Trait Locus: A Random Model Approach

Dear Dr Xu,

I am pleased to inform you that your manuscript has been formally accepted for publication in PLOS Computational Biology. Your manuscript is now with our production department and you will be notified of the publication date in due course.

With kind regards,

Orsolya Voros
